# ONE-CLASS CLASSIFICATION ROBUST TO GEOMETRIC TRANSFORMATIONS

## ABSTRACT

Recent studies on one-class classification have achieved a remarkable performance, by employing the self-supervised classifier that predicts the geometric transformation applied to in-class images. However, they cannot identify in-class images at all when the input images are geometrically-transformed (e.g., rotated images), because their classification-based in-class scores assume that input images always have a fixed viewpoint, as similar to the images used for training. Pointing out that humans can easily recognize such transformed images as the same class, in this work, we aim to propose a one-class classifier robust to geometrically-transformed inputs, named as GROC. To this end, we introduce a conformity score which indicates how strongly an input image agrees with one of the predefined in-class transformations, then utilize the conformity score with our proposed agreement measures for one-class classification. Our extensive experiments demonstrate that GROC is able to accurately distinguish in-class images from out-of-class images regardless of whether the inputs are geometrically-transformed or not, whereas the existing methods fail.

## 1 INTRODUCTION

One-class classification refers to the problem of identifying whether an input example belongs to a single target class (*in-class*) or any of novel classes (*out-of-class*). The main challenge of this task is that only in-class examples are available at training time. Thus, by using only positive examples, a model has to learn the decision boundary that distinguishes in-class examples from out-of-class examples, whose distribution is assumed to be unknown in practice. Early work on one-class classification mainly utilized kernel-based methods (Schölkopf et al., 2000; Tax & Duin, 2004) to find a hypersphere (or hyperplane) enclosing all training in-class examples, or density estimation techniques (Parzen, 1962) to measure the likelihood of an input example.

In the era of deep learning, numerous literature have tried to employ deep neural networks to effectively learn the high-dimensional data (e.g., images). Most of them aim to detect out-of-class examples based on density estimation, by adopting the architecture of autoencoders (Ruff et al., 2018; Zong et al., 2018) or generative adversarial networks (GANs) (Schlegl et al., 2017; Zenati et al., 2018). Nevertheless, their supervision is not useful enough to capture the semantic of high-dimensional data for a target class, which eventually leads to the limited performance. Recently, there have been several attempts to make use of self-supervised learning (Golan & El-Yaniv, 2018; Hendrycks et al., 2019; Bergman & Hoshen, 2020) for more informative supervision on the target class, and made a major breakthrough to this problem. They build a self-labeled image set by applying a bunch of geometric transformations to training images, then train a classifier to accurately predict the transformation applied to original input images. This approach achieved the state-of-the-art performance for one-class classification even without modeling the latent distribution of in-class examples for density estimation.

However, all the aforementioned methods are quite vulnerable to spatial variances within the images, because they were developed based on the assumption that in-class (and out-of-class) images have a fixed viewpoint. In particular, the existing self-supervised methods do not work completely for the inputs with various viewpoints in that their capability of predicting the geometric transformation relies on the fixed viewpoint. Note that humans usually recognize that the images of a target object with different viewpoints belong to the same class; in this sense, the one-class classifiers also

should be robust to the viewpoint of input images. In other words, we need to make geometrically-transformed in-class images not to be identified as out-of-class, from the perspective that a geometric transformation (e.g., rotation & x,y-translation) does not change the semantic (i.e., object class) but the viewpoint.

The goal of our work is to propose an effective strategy that can circumvent the limitation of viewpoint sensitivity, without compromising the performance for the images with the fixed viewpoint. We first present several evaluation settings for validating the robustness to flexible viewpoints, artificially introduced by geometric transformations. Then, we describe our proposed solution, termed as GROC, which measures a *conformity score* indicating how confidently an input image matches with one of the predefined (*anchor*) in-class transformations. In this work, we offer two measures for the conformity score, which are the inner product similarity and the conditional likelihood, and show how they can be optimized by the training in-class images. The empirical experiments on the proposed evaluation scenarios show that GROC considerably outperforms all the other competing methods in terms of the robustness to geometric transformation.

## 2 PRELIMINARIES

### 2.1 PROBLEM FORMULATION

Let $\mathcal{X}$ be a set of all kinds of images, $\mathcal{X}_{in} \subseteq \mathcal{X}$ and $\mathcal{X}_{out} = \mathcal{X} \backslash \mathcal{X}_{in}$ be the sets of all *in-class* and *out-of-class* images, respectively. Given training in-class data $\mathcal{X}_{in}^{tr} \subseteq \mathcal{X}_{in}$, we consider the one-class classification problem which differentiates in-class and out-of-class data. The problem aims to build a classifier by using only the known in-class data for training. The classifier learns an in-class score function, $S_{in}(\mathbf{x}) : \mathcal{X} \to \mathbb{R}$, where a higher score indicates that the input $\mathbf{x}$ is more likely to be in $\mathcal{X}_{in}$. Based on the score, the classifier determines whether the input belongs to in-class or not.

### 2.2 SELF-SUPERVISED LEARNING METHODS FOR ONE-CLASS CLASSIFICATION

Recently, the self-supervised learning methods (Golan & El-Yaniv, 2018; Hendrycks et al., 2019; Bergman & Hoshen, 2020) have achieved the state-of-the-art performance in one-class classification. For self-supervised learning, they first create a self-labeled dataset and use it to train a multi-class classifier. Concretely, let $\mathcal{T} = \{T_0, \cdots, T_i, \cdots, T_{K-1}\}$ be a set of predefined (*anchor*) geometric transformations, where $T_0(x) = x$ is the identity mapping and each transformation $T_i$ is a composition of multiple unit transformations (i.e., rotation & x,y-translation). The self-labeled dataset consists of transformed images and their corresponding labels.

$$\mathcal{D}_{self} = \{(T_i(\mathbf{x}), i) | \mathbf{x} \in \mathcal{X}_{in}^{tr}, 0 \le i < K\}, \tag{1}$$

where $T_i(\cdot)$ is the $i$-th transformation operator and its label $i$ is the transformation id of $T_i(\cdot)$.

Using the self-labeled dataset, these methods train a softmax classifier based on a multi-class classification loss (i.e., cross-entropy) for a discrimination among the transformations. For one-class classification, they define an in-class score under the assumption that a well-trained classifier would better predict the transformation for the in-class images than that for the out-of-class images. In the end, the in-class score for an unseen image $\mathbf{x}$ is defined by the sum of softmax probabilities that its transformed images are correctly classified as their labels (Golan & El-Yaniv, 2018; Bergman & Hoshen, 2020).

$$S_{in}(\mathbf{x}) = \sum_{i=0}^{K-1} p\left(y = i | T_i(\mathbf{x})\right), \tag{2}$$

where $p\left(y = i | T_i(\mathbf{x})\right)$ is the softmax probability that $T_i(\mathbf{x})$ is classified as the $i$-th transformation.

The state-of-the-art method based on this self-supervised approach (Hendrycks et al., 2019) significantly improves the performance by formulating the classification task in a multi-label manner. Since each transformation is determined by the combination of unit transformations from three categories[1], (i.e., rotation, (horizontal) x-translation, and (vertical) y-translation), the unit transformations applied to an input image can be independently predicted for each category. Thus, they adopt a

---

[1]They build the set of transformations $\mathcal{T}$ by the combination of the following unit transformations: rotation $\in \{0°, 90°, 180°, 270°\}$, x-translation $\in \{-8, 0, +8\}$, and y-translation $\in \{-8, 0, +8\}$.

softmax head for each transformation category, then train a classifier to predict the degree of transformations within each category. The final in-class score is also replaced with the one summarizing all the softmax heads, each of which is for the unit transformation applied to the input.

# 3 METHOD

## 3.1 MOTIVATION

The underlying concept of the self-supervised methods based on transformation classification is to learn discriminative features of in-class images, in order to classify various viewpoints caused by the geometric transformations. The precondition for this approach is that the viewpoint of training images is always the same, otherwise the classifier cannot be trained due to the inconsistent supervision. However, at test time, the input images can have different viewpoints from those appearing in the training images. We remark that the images of the same object with different viewpoints belong to the same class, as usually recognized by humans. In this sense, it is desired that in-class images with various viewpoints are identified as in-class, not out-of-class. That is, the robustness to geometric transformations should be considered for one-class classification.

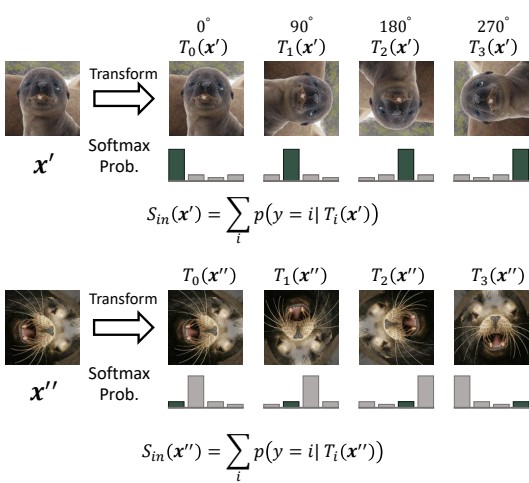

In this respect, the existing self-supervised methods totally fail to compute the effective in-class scores for inputs with various viewpoints. We observe that they produce an undesirable in-class score especially when the input image has the same (or similar) viewpoint represented by the anchor transformations $\mathcal{T} \backslash \{T_0\}$. For example, suppose a classifier is trained on $\mathcal{D}_{self}$ with the transformations $\mathcal{T}$ of clockwise rotations $\{0°, 90°, 180°, 270°\}$. Given two images of sea lions $\mathbf{x}'$ and $\mathbf{x}''$, let $\mathbf{x}'$ have the same viewpoint with the training images and $\mathbf{x}''$ have the $90°$ rotated viewpoint, which is equivalent to $T_1(\mathbf{x}')$. As illustrated in Figure 1, the softmax probability of each transformed image has a high value for the input $\mathbf{x}'$, but the one for $\mathbf{x}''$ has a low value. Consequently, it cannot correctly identify $\mathbf{x}''$ as in-class, though it comes from the target class as well. We point out that setting the target label of each transformed image to the applied transformation is not valid any longer when an input viewpoint is changed.

Figure 1: A toy example of computing the in-class scores for two in-class (sea lion) images with different viewpoints. Each row represents how the in-class score is calculated for a given input.

A straightforward solution for this challenge is augmenting the training dataset so that it can cover various viewpoints of in-class images. Unfortunately, the data augmentation technique is not applicable because it results in inconsistent supervision for the task of discriminating the viewpoints, which is the learning objective of the self-supervised methods. On the other hand, there exist several one-class classification methods (Ruff et al., 2018; Zong et al., 2018) that can adopt the data augmentation technique. However, they cannot achieve the performance as high as the self-supervised methods even in the case that all input images have a fixed viewpoint; this will be further discussed in Section 4. To sum up, we need to consider another strategy to develop a robust one-class classifier that works well even for the input images having various viewpoints.

## 3.2 PROPOSED SETUPS

We first propose three evaluation setups for testing the robustness to various viewpoints: 1) fixed viewpoint, 2) anchor viewpoint, and 3) random viewpoint. We artificially introduce the spatial variance (i.e., the changes of the viewpoint) in test images by using the geometric transformations. Note that $\mathcal{X}^{te}$ denotes the test data, which contains both in-class and out-of-class images.

**Fixed viewpoint setup.** In this setup, we consider only the fixed viewpoint that is used for training, as done in previous work. We do not change the viewpoint of the original test images, $\mathcal{X}^{te}_{fv} = \mathcal{X}^{te}$.

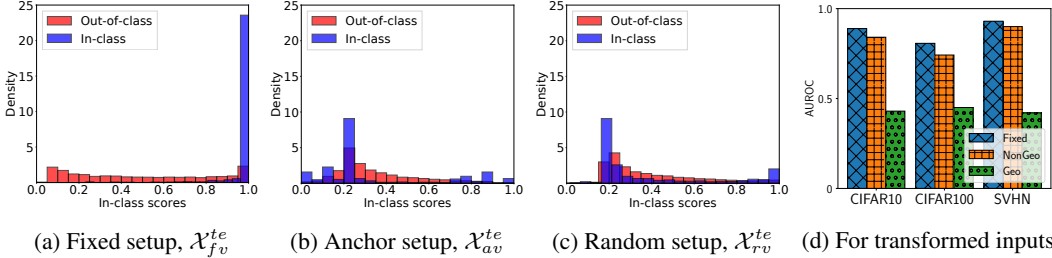

(a) Fixed setup, $\mathcal{X}_{fv}^{te}$    (b) Anchor setup, $\mathcal{X}_{av}^{te}$    (c) Random setup, $\mathcal{X}_{rv}^{te}$    (d) For transformed inputs

Figure 2: The one-class classification performance of the self-supervised method (Hendrycks et al., 2019). (a-c) The in-class score distributions of in-class and out-of-class test images (Dataset: CIFAR-10, In-class: Horse), (d) AUROC for geometrically/non-geometrically transformed inputs.

**Anchor viewpoint setup.** This setup is designed for verifying the robustness to the viewpoints induced by the anchor transformations. We build a test dataset by $\mathcal{X}_{av}^{te} = \{T(\mathbf{x})|T \sim \mathcal{T}, \mathbf{x} \in \mathcal{X}^{te}\}$, where $T$ is randomly sampled from the set of the anchor transformations $\mathcal{T}$ for each image $\mathbf{x}$.

**Random viewpoint setup.** The random viewpoint setup further considers the geometric transformations that are not included in the set of anchor transformations. We first define the superset of $\mathcal{T}$, denoted by $\mathcal{T}^*$, including a number of transformations with continuous degrees. A test dataset for this setup is built by $\mathcal{X}_{rv}^{te} = \{T(\mathbf{x})|T \sim \mathcal{T}^*, \mathbf{x} \in \mathcal{X}^{te}\}$, where $T$ is sampled for each image $\mathbf{x}$.

As a preliminary result, we plot the in-class score distributions for in-class and out-of-class test images, computed by the state-of-the-art self-supervised method (Hendrycks et al., 2019). In Figure 2, we observe that the score distributions of in-class and out-of-class images in $\mathcal{X}_{fv}^{te}$ are clearly distinguishable, which supports the great performance for one-class classification. On the contrary, the two score distributions are almost overlapping with each other in cases of $\mathcal{X}_{av}^{te}$ and $\mathcal{X}_{rv}^{te}$, strongly indicating that they fail to figure out in-class images due to their various viewpoints.

We additionally investigate the performance drop for geometrically/non-geometrically transformed inputs. In Figure 2(d), it is obvious that geometric transformations make the self-supervised method totally malfunction, while non-geometric transformations (e.g., brightness, contrast, sharpness, and color temperature) hardly degrade the final performance for one-class classification.

### 3.3 Proposed Strategy

To deal with the viewpoint sensitivity of the self-supervised methods, we note that the in-class images match better with the in-class transformations than the out-of-class images do, regardless of their viewpoints. Our proposed strategy, named as GROC, defines the in-class score by the sum of the *conformity* scores for $K$ transformed images; $S_{conf}(\cdot; \mathcal{T})$ calculates how conformable an input image is to the given set $\mathcal{T}$. Formally, it is defined by the maximum similarity between the representation of an input image and that of each anchor transformation:

$$S_{in}(\mathbf{x}) = \sum_{i=0}^{K-1} S_{conf}(T_i(\mathbf{x}); \mathcal{T}), \text{ where } S_{conf}(\mathbf{x}; \mathcal{T}) = \max_{T_j \in \mathcal{T}} \left[ sim\left(\mathbf{x}, T_j\right) \right]. \tag{3}$$

The foremost condition for GROC is that the representations of the anchor transformations should be discriminative, so that the similarity measure can effectively capture the viewpoint of input images. Note that the similarity between an image $\mathbf{x}$ and a transformation $T_j$, denoted by $sim(\mathbf{x}, T_j)$, can be defined in various ways. In the following subsections, we offer two similarity measures for the conformity score, respectively modeled by inner product similarity and conditional likelihood, and present how the representations of input images and anchor transformations are optimized.

#### 3.3.1 Inner Product Similarity for Conformity Score

To model the similarity measure for the conformity score, we use our encoder network $f(\cdot; \boldsymbol{\theta})$ : $\mathcal{X} \to \mathbb{R}^d$ which outputs the representation of an input image, and the weight matrix $\boldsymbol{W} \in \mathbb{R}^{K \times d}$ that parameterizes the representations of $K$ anchor transformations. We first present GROC-IP whose similarity measure is simply computed by the inner product of $f(\mathbf{x}; \boldsymbol{\theta})$ and $\mathbf{w}_j$:

$$sim\left(\mathbf{x}, T_j\right) = f(\mathbf{x}; \boldsymbol{\theta})^\top \mathbf{w}_j. \tag{4}$$

Based on the inner product similarity, the encoder network needs to map all in-class images with the same viewpoint close to their corresponding transformation vector, while keeping the transformation vectors far from each other. In other words, it has to extract discriminative features for classifying the input images according to their viewpoint; this has been already achieved by the conventional softmax classifier in a self-supervised manner.

Therefore, we adopt the optimization strategy provided by the existing self-supervised methods for one-class classification. After we build a softmax classifier by adding the linear classification layer of weights $\boldsymbol{W}$ on the top of the encoder network $f(\cdot; \boldsymbol{\theta})$, we train both the weight matrix and the network by using the cross-entropy loss with $\mathcal{D}_{self}$. In case of GROC-IP, the conformity score becomes equivalent to the maximum logit value computed by the softmax classifier.

### 3.3.2 CONDITIONAL LIKELIHOOD FOR CONFORMITY SCORE

Our second method, GROC-CL, defines the similarity measure by using the likelihood of an input image conditioned on each transformation. For simplicity, we assume that the conditional likelihood has the form of an isotropic Gaussian distribution, whose mean is $\boldsymbol{\mu}_j \in \mathbb{R}^d$ and standard deviation is $\sigma_j \in \mathbb{R}_+$ for the condition $j$ (i.e., transformation $T_j$). In short, GROC-CL models the representation of $T_j$ as $(\boldsymbol{\mu}_j, \sigma_j)$ rather than $\mathbf{w}_j$. Similar to Section 3.3.1, all the Gaussian distributions need to be separable for a discrimination among different viewpoints. Based on this assumption, the similarity between $\mathbf{x}$ and $T_j$ is defined by the log-likelihood as follows.

$$ sim\left(\mathbf{x}, T_j\right) = \log \mathcal{N}(f(\mathbf{x}; \boldsymbol{\theta}) | \boldsymbol{\mu}_j, \sigma_j^2 I) \approx - \left( \frac{\|f(\mathbf{x}; \boldsymbol{\theta}) - \boldsymbol{\mu}_j\|_2^2}{2\sigma_j^2} + \log \sigma_j^d \right). \tag{5} $$

Note that this similarity can be interpreted as the Mahalanobis distance between $f(\mathbf{x}; \boldsymbol{\theta})$ and $\boldsymbol{\mu}_j$ with the covariance matrix $\sigma_j^2 I$. The challenge here is to optimize the encoder network so that its outputs follow $\mathcal{N}(\boldsymbol{\mu}_j, \sigma_j^2 I)$ for all in-class inputs having the viewpoint corresponding to $T_j$. We train the parameters for the Gaussian distributions $(\boldsymbol{\mu}, \boldsymbol{\sigma})$ and the network $f(\cdot; \boldsymbol{\theta})$ by the following objective.

$$ \max_{\boldsymbol{\theta}, \boldsymbol{\mu}, \boldsymbol{\sigma}, \boldsymbol{b}} \sum_{\mathbf{x} \in \mathcal{X}_{in}^{tr}} \sum_{j=0}^{K-1} \left[ sim\left(T_j(\mathbf{x}), T_j\right) + \frac{1}{\nu} \log \frac{\exp\left(sim\left(T_j(\mathbf{x}), T_j\right) + b_j\right)}{\sum_{k=0}^{K-1} \exp\left(sim\left(T_j(\mathbf{x}), T_k\right) + b_k\right)} \right], \tag{6} $$

where $b_j \in \mathbb{R}$ is the bias term for transformation $j$. As discussed in (Lee et al., 2020), this objective aims to learn the discriminative conditional likelihoods modeled by $K$ separable Gaussian distributions. To be precise, the first term is enforcing that $f(T_j(\mathbf{x}))$ follows the $j$-th Gaussian distribution, and the second term makes them distinguishable based on Gaussian discriminant analysis.

## 4 EXPERIMENTS

### 4.1 COMPETING METHODS

In our experiments, we consider a variety of approaches to one-class classification as the competing methods. We choose One-class SVM (**OCSVM**) (Schölkopf et al., 2000) and Deep SVDD (**DSVDD**) (Ruff et al., 2018) as the non-self-supervised methods. OCSVM is a classical kernel-based method for one-class classification, which finds a maximum-margin hyperplane the separates enclosing most of the training in-class examples. DSVDD, a deep learning variant of OCSVM, explicitly models the latent space in which training in-class examples gather to a specific center point.

The main competitors are the self-supervised methods based on transformation classification: Geometric Transformation (**GT**) (Golan & El-Yaniv, 2018) and Multi-labeled Geometric Transformation (**MGT**) (Hendrycks et al., 2019). The details of the methods are presented in Section 2.2. For the anchor transformations, GT adopts four transformation categories (i.e., horizontal flipping, x,y-translation, and rotation), while MGT excludes horizontal flipping from the above categories.

The last competing method is **SimCLR** (Chen et al., 2020) which learns the transformation-invariant representations of input images in a self-supervised manner. Since it is optimized to maximize the agreement among the images differently-transformed from a single image, it is capable of alleviating

the viewpoint sensitivity to some degree. Several recent work on representation learning based on this approach (Chen et al., 2020; Grill et al., 2020; He et al., 2020) showed the remarkable performance for a wide range of downstream tasks. Note that SimCLR is not originally designed for one-class classification, thus we tailor it for our task. We define the final in-class score by

$$S_{in}(\mathbf{x}) = \sum_{i=1}^{K-1} \frac{f(T_0(\mathbf{x}); \boldsymbol{\theta})^\top f(T_i(\mathbf{x}); \boldsymbol{\theta})}{\|f(T_0(\mathbf{x}); \boldsymbol{\theta})\|_2 \|f(T_i(\mathbf{x}); \boldsymbol{\theta})\|_2}. \tag{7}$$

For its optimization, we use the set of the anchor transformations adopted by MGT. More details of SimCLR are provided in Appendix A.

## 4.2 DATASETS

We validate the effectiveness of the proposed methods using three benchmark image datasets: CIFAR-10, CIFAR-100 (Krizhevsky et al., 2009), and SVHN (Netzer et al., 2011). We scale the pixel values of all images to be in $[-1, 1]$ as done in (Golan & El-Yaniv, 2018). Note that CIFAR-100 has 20 super-classes and we use these super-classes rather than 100 full classes.

## 4.3 EXPERIMENTAL SETTINGS

Following the experimental setting of the previous studies (Golan & El-Yaniv, 2018; Ruff et al., 2018; Zenati et al., 2018), we employ the one-vs-all evaluation scheme. For the dataset with $C$ classes, we generate $C$ one-class classification settings; the images of a target class are regarded as in-class data, and the other images belonging to $C - 1$ classes are regarded as out-of-class data. We use the Area Under the ROC curve (AUROC) as an evaluation metric.

We build the set of the anchor transformations by the combination of the following unit transformations: x-translation $\in \{-8, 0, +8\}$, y-translation $\in \{-8, 0, +8\}$, and rotation $\in \{0°, 90°, 180°, 270°\}$. As presented in Section 3.2, we evaluate our method by using the three proposed setups: fixed viewpoint, anchor viewpoint, and random viewpoint. For the random viewpoint setup, we build the test set by randomly sampling the transformation degrees for the x,y-translation in the range of $[-8, 8]$, and for the rotation in the range of $[0°, 360°]$. In case of SVHN, we exclude the x,y-translation because it may change the semantic (i.e., class label) of the images; the label of each image is determined by the digit in the middle of the image.

## 4.4 EXPERIMENTAL RESULTS

### 4.4.1 COMPARISONS WITH SELF-SUPERVISED METHODS

The experimental results of the self-supervised methods on the three setups are presented in Table 1. For each dataset, the original class name is replaced by its class id due to the limited space. In summary, our methods effectively overcome the limitation of viewpoint sensitivity, without compromising the performance in the fixed viewpoint setup. We analyze the results from various perspectives.

**Fixed viewpoint setup.** In this setup, the state-of-the-art competitor MGT consistently shows the best results, and our methods show the comparable results to MGT. We also observe that the performance of SimCLR is not as good as those of the classification-based methods. Note that the classification-based methods aim for a discrimination among the differently-transformed images, whereas SimCLR tries to make them indistinguishable. From this observation, we can conclude that directly learning the transformation-invariant visual features is less effective to identify in-class and out-of-class images for one-class classification.

**Anchor viewpoint setup.** In case that the input images have anchor viewpoints, the classification-based methods fail to distinguish in-class images from out-of-class images; their performances are even worse than a random guess whose AUROC is 0.5. Because their capability of discriminating the geometric transformations depends on the fixed viewpoint, they do not work at all for the input images with various viewpoints, as discussed in Section 3.1. In contrast, our methods considerably outperform all the competing methods while providing outstanding performances robust to the changes of the viewpoint. These results show that our methods successfully identify the in-class images irrespective of their viewpoint, by the help of the conformity score that measures how

Table 1: The AUROC of self-supervised methods for one-class classification on the three evaluation setups. The best results are marked in bold face.

| Dataset | in-class | Fixed Viewpoint | Anchor Viewpoint | Random Viewpoint |
|---|---|---|---|---|
| | | SimCLR / GT / MGT / GROC-IP/ GROC-CL | | |
| CIFAR-10 | 0 | 0.66 / 0.77 / **0.78** / 0.76 / 0.75 | 0.54 / 0.44 / 0.45 / **0.70** / 0.68 | 0.53 / 0.44 / 0.44 / 0.63 / **0.64** |
| | 1 | 0.62 / 0.87 / 0.96 / 0.95 / **0.97** | 0.40 / 0.32 / 0.39 / 0.91 / **0.94** | 0.40 / 0.39 / 0.40 / 0.84 / **0.88** |
| | 2 | 0.64 / 0.83 / **0.86** / 0.85 / 0.84 | 0.57 / 0.42 / 0.45 / **0.76** / 0.73 | 0.56 / 0.46 / 0.45 / 0.64 / **0.65** |
| | 3 | 0.54 / 0.78 / **0.80** / 0.77 / 0.77 | 0.43 / 0.45 / 0.44 / **0.73** / 0.69 | 0.43 / 0.47 / 0.49 / **0.62** / 0.61 |
| | 4 | 0.61 / 0.86 / **0.91** / 0.90 / 0.90 | 0.49 / 0.39 / 0.42 / **0.85** / 0.83 | 0.49 / 0.38 / 0.42 / **0.72** / 0.71 |
| | 5 | 0.54 / 0.87 / **0.89** / 0.88 / 0.88 | 0.43 / 0.42 / 0.42 / **0.83** / **0.83** | 0.44 / 0.44 / 0.44 / 0.72 / **0.73** |
| | 6 | 0.65 / 0.89 / 0.89 / 0.90 / **0.93** | 0.42 / 0.44 / 0.42 / 0.81 / **0.83** | 0.42 / 0.44 / 0.46 / 0.72 / **0.76** |
| | 7 | 0.58 / 0.91 / **0.96** / 0.95 / **0.96** | 0.43 / 0.32 / 0.42 / **0.91** / 0.89 | 0.43 / 0.40 / 0.40 / **0.77** / **0.77** |
| | 8 | 0.52 / 0.88 / **0.94** / 0.92 / **0.94** | 0.44 / 0.35 / 0.41 / **0.88** / 0.87 | 0.41 / 0.36 / 0.39 / **0.80** / **0.80** |
| | 9 | 0.51 / 0.84 / 0.90 / 0.90 / **0.93** | 0.41 / 0.36 / 0.42 / 0.86 / **0.88** | 0.40 / 0.39 / 0.40 / 0.79 / **0.83** |
| | avg | 0.59 / 0.85 / **0.89** / 0.88 / **0.89** | 0.46 / 0.39 / 0.42 / **0.82** / **0.82** | 0.45 / 0.42 / 0.43 / 0.73 / **0.74** |
| CIFAR-100 | 0 | 0.57 / 0.75 / 0.77 / 0.77 / **0.78** | 0.55 / 0.43 / 0.43 / 0.71 / **0.72** | 0.55 / 0.44 / 0.47 / 0.62 / **0.64** |
| | 1 | 0.64 / 0.69 / 0.72 / **0.74** / 0.72 | 0.58 / 0.46 / 0.43 / **0.66** / 0.62 | 0.60 / 0.48 / 0.47 / 0.60 / **0.62** |
| | 2 | 0.63 / 0.71 / 0.71 / 0.75 / **0.82** | 0.49 / 0.44 / 0.48 / 0.70 / **0.75** | 0.51 / 0.48 / 0.47 / 0.65 / **0.70** |
| | 3 | 0.58 / 0.78 / **0.80** / **0.80** / **0.80** | 0.55 / 0.40 / 0.41 / 0.72 / **0.75** | 0.50 / 0.48 / 0.47 / 0.52 / **0.65** |
| | 4 | 0.63 / 0.78 / 0.80 / **0.81** / **0.81** | 0.53 / 0.42 / 0.46 / **0.76** / **0.76** | 0.50 / 0.46 / 0.43 / 0.70 / **0.72** |
| | 5 | 0.59 / 0.66 / **0.67** / **0.67** / 0.66 | 0.56 / 0.45 / 0.49 / **0.60** / 0.59 | 0.53 / 0.48 / 0.51 / 0.53 / **0.55** |
| | 6 | 0.50 / 0.86 / **0.88** / 0.84 / 0.85 | 0.48 / 0.43 / 0.46 / 0.77 / **0.82** | 0.44 / 0.46 / 0.46 / **0.59** / 0.58 |
| | 7 | 0.67 / 0.65 / 0.66 / 0.67 / **0.69** | 0.56 / 0.46 / 0.47 / 0.60 / **0.62** | 0.57 / 0.48 / 0.48 / **0.59** / 0.56 |
| | 8 | 0.50 / 0.84 / **0.87** / 0.85 / **0.87** | 0.48 / 0.43 / 0.46 / 0.78 / **0.80** | 0.45 / 0.47 / 0.46 / 0.69 / **0.75** |
| | 9 | 0.42 / 0.88 / **0.92** / 0.91 / 0.91 | 0.41 / 0.39 / 0.41 / **0.89** / 0.87 | 0.41 / 0.38 / 0.39 / 0.76 / **0.77** |
| | 10 | 0.60 / 0.85 / **0.87** / 0.85 / 0.85 | 0.54 / 0.39 / 0.44 / **0.82** / 0.81 | 0.56 / 0.42 / 0.43 / 0.73 / **0.74** |
| | 11 | 0.53 / 0.84 / 0.86 / 0.86 / **0.87** | 0.46 / 0.41 / 0.43 / 0.80 / **0.81** | 0.48 / 0.47 / 0.44 / **0.70** / **0.70** |
| | 12 | 0.54 / 0.82 / **0.84** / 0.83 / 0.84 | 0.47 / 0.43 / 0.45 / 0.77 / **0.80** | 0.47 / 0.45 / 0.45 / 0.68 / **0.72** |
| | 13 | **0.65** / 0.60 / 0.62 / 0.60 / 0.59 | **0.63** / 0.51 / 0.49 / 0.56 / 0.55 | **0.61** / 0.49 / 0.47 / 0.51 / 0.52 |
| | 14 | 0.52 / 0.92 / **0.93** / 0.90 / 0.90 | 0.44 / 0.40 / 0.42 / **0.86** / 0.84 | 0.44 / 0.42 / 0.43 / **0.73** / 0.70 |
| | 15 | 0.56 / 0.69 / 0.70 / 0.71 / **0.72** | 0.55 / 0.45 / 0.46 / **0.65** / **0.65** | 0.53 / 0.47 / 0.49 / 0.60 / **0.61** |
| | 16 | 0.55 / 0.76 / **0.80** / **0.80** / 0.76 | 0.52 / 0.48 / 0.43 / **0.73** / 0.68 | 0.52 / 0.49 / 0.46 / **0.66** / **0.66** |
| | 17 | 0.65 / 0.90 / **0.93** / 0.92 / 0.92 | 0.55 / 0.36 / 0.41 / **0.88** / 0.84 | 0.52 / 0.38 / 0.39 / **0.79** / 0.76 |
| | 18 | 0.52 / 0.89 / 0.90 / 0.90 / **0.92** | 0.44 / 0.40 / 0.40 / **0.87** / 0.86 | 0.41 / 0.47 / 0.42 / 0.75 / **0.76** |
| | 19 | 0.53 / 0.85 / **0.89** / 0.87 / 0.85 | 0.45 / 0.40 / 0.41 / **0.81** / 0.79 | 0.47 / 0.43 / 0.39 / **0.73** / 0.71 |
| | avg | 0.57 / 0.79 / **0.81** / 0.80 / **0.81** | 0.51 / 0.43 / 0.44 / **0.75** / **0.75** | 0.50 / 0.45 / 0.45 / 0.66 / **0.67** |
| SVHN | 0 | 0.79 / 0.71 / 0.86 / **0.92** / **0.92** | 0.79 / 0.39 / 0.21 / **0.92** / **0.92** | 0.63 / 0.43 / 0.44 / 0.78 / **0.80** |
| | 1 | 0.68 / 0.72 / 0.80 / 0.89 / **0.90** | 0.68 / 0.35 / 0.20 / **0.89** / **0.89** | 0.52 / 0.42 / 0.43 / 0.75 / **0.77** |
| | 2 | 0.75 / **0.98** / 0.96 / 0.94 / 0.94 | 0.75 / 0.30 / 0.23 / **0.94** / **0.94** | 0.53 / 0.40 / 0.41 / 0.75 / **0.76** |
| | 3 | 0.63 / 0.91 / **0.94** / 0.93 / 0.92 | 0.63 / 0.35 / 0.41 / **0.93** / 0.91 | 0.60 / 0.42 / 0.42 / **0.75** / 0.73 |
| | 4 | 0.63 / **0.98** / **0.98** / 0.95 / 0.96 | 0.63 / 0.33 / 0.21 / 0.95 / **0.96** | 0.54 / 0.42 / 0.42 / **0.71** / **0.71** |
| | 5 | 0.86 / **0.96** / 0.95 / 0.95 / 0.94 | 0.86 / 0.30 / 0.32 / **0.95** / 0.94 | 0.69 / 0.39 / 0.40 / **0.83** / 0.78 |
| | 6 | 0.69 / **0.96** / **0.96** / 0.91 / 0.91 | 0.69 / 0.31 / 0.35 / **0.91** / **0.91** | 0.61 / 0.36 / 0.39 / **0.80** / 0.78 |
| | 7 | 0.75 / **0.99** / 0.98 / 0.96 / 0.96 | 0.75 / 0.30 / 0.23 / **0.96** / **0.96** | 0.46 / 0.39 / 0.43 / 0.70 / **0.74** |
| | 8 | 0.67 / 0.77 / 0.88 / **0.92** / 0.91 | 0.67 / 0.36 / 0.33 / **0.92** / 0.91 | 0.59 / 0.44 / 0.46 / **0.71** / 0.67 |
| | 9 | 0.65 / **0.97** / **0.97** / 0.90 / 0.89 | 0.65 / 0.32 / 0.31 / **0.90** / 0.89 | 0.60 / 0.34 / 0.43 / **0.79** / 0.78 |
| | avg | 0.71 / 0.90 / **0.93** / **0.93** / 0.92 | 0.71 / 0.33 / 0.28 / **0.93** / 0.92 | 0.58 / 0.40 / 0.42 / **0.76** / 0.75 |

confidently an input image matches with one of the in-class transformations. Interestingly, the performance of SimCLR is higher than that of the classification-based methods in this setup. This is because its learning objective that encourages the multiple transformations of an input image to be similar with each other makes the one-class classifier less affected by the viewpoint.

**Random viewpoint setup.** In the hardest setting where the input images have random viewpoints, the classification-based methods cannot beat a random guessing, similarly to the anchor viewpoint setup. Our methods perform the best for all the datasets, which strongly indicates the robustness to the changes of the viewpoint. In conclusion, both of the our methods (i.e., GROC-IP and GROC-CL) are able to correctly classify the images having diverse viewpoints into in-class or out-of-class, even for the viewpoints that have not been seen during the training.

### 4.4.2 COMPARISONS WITH NON-SELF-SUPERVISED METHODS

Figure 3 presents the comparison results against the non-self-supervised methods on the three evaluation setups. Due to the limited space, we report the score averaged over $C$ in-class settings for

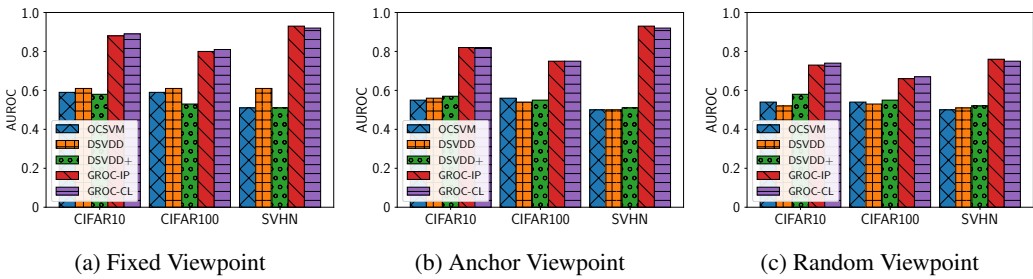

(a) Fixed Viewpoint  (b) Anchor Viewpoint  (c) Random Viewpoint

Figure 3: Comparisons with non-self-supervised methods in the three evaluation setups

each dataset. We also provide the results of DSVDD that adopts the data augmentation technique, denoted by **DSVDD+**; the training in-class images are randomly augmented by $T \sim \mathcal{T}^*$. For all the setups, the competing methods show poorer performances compared to our methods. Specifically, the data augmentation technique results in the limited performance gains in the anchor/random setups, whereas it even brings an adverse effect in the fixed setup. This implies that the simple approach is not sufficient to address the viewpoint sensitivity of the one-class classifiers.

### 4.4.3 FURTHER ANALYSIS

We also provide in-depth analyses on the performance of the self-supervised methods, for the SVHN dataset. In the fixed viewpoint setup, we observe that there is a distinct performance improvement of our GROC over MGT, especially for the cases of in-class 0, 1, and 8. Figure 4 shows the in-class score distributions obtained by GROC and MGT, where the class 0 is set to in-class. Since the digit '0' has a symmetric shape, it is difficult for MGT to differentiate its transformations between the rotation $0°$ and $180°$ (or, $90°$ and $270°$). For this reason, as illustrated in the rightmost figure, MGT outputs relatively low scores for the images having a single '0' but high scores for the images having other digits around '0'. On the contrary, GROC produces the similar (and high) in-class scores for these images (i.e., with or without other digits), which can be separated from the scores of out-of-class images. This helps GROC to less overlap the in-class scores between in-class and out-of-class images, and as a result, it leads to higher AUROC compared to MGT.

On the other hand, for the cases of in-class 6 and 9, the performance of GROC slightly degrades because the $180°$ rotation of the out-of-class digit '9' is more likely to be conformable to the in-class digit '6', and vice versa. Nevertheless, under the assumption that the input images have various viewpoints, it is impossible even for humans to accurately figure out whether the images that look like '6' (or '9') belong to in-class or out-of-class.

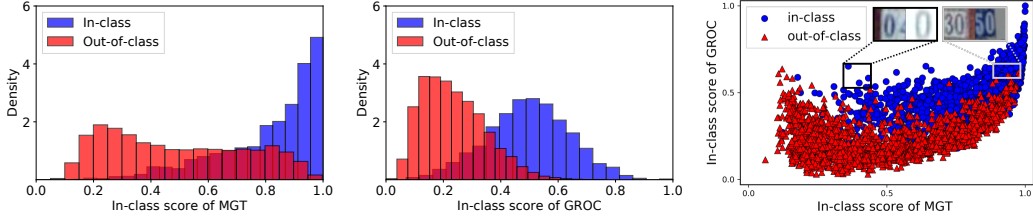

Figure 4: The in-class score distributions of in-class and out-of-class test images in the fixed viewpoint setup (Dataset: SVHN, In-class: 0).

## 5 CONCLUSION

This paper proposes a novel one-class classification method robust to geometric transformations, which effectively addresses the challenge that in-class images cannot be correctly distinguished from out-of-class images when they have various viewpoints. We first present new evaluation setups that cover the diverse viewpoints by artificially introducing the spatial variance into test images. Then, we define the conformity-based in-class score so as to measure how strongly an input image is conformable to one of the anchor transformations, whose representations are optimized to be discriminative. The extensive experiments demonstrate that the proposed GROC keeps its outstand-

ing performance even in the anchor/random viewpoint setups where the input images have various viewpoints, whereas the state-of-the-art methods perform even worse than a random guessing.

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

## A  SIMCLR

In the experiments, we slightly modified SimCLR (Chen et al., 2020) for the one-class classification task. The main idea of SimCLR is learning representations by maximizing the agreement among the images differently-transformed from the same image via a contrastive loss in the latent space. For the optimization of SimCLR, we use the set of the anchor transformations $\mathcal{T}$ that is adopted by the classification-based self-supervised method (i.e., MGT); this set is different from the one used by the original SimCLR. Let $\mathbf{x} \in \mathcal{X}_{in}^{tr}$ and $\tilde{\mathbf{x}} = T(\mathbf{x})$ where $T$ is a transformation operator randomly sampled from the set of the anchor transformations $\mathcal{T}$. Given a batch $\mathcal{B} = \{\mathbf{x}_1, \cdots, \mathbf{x}_N\} \subset \mathcal{X}_{in}^{tr}$, we define $\tilde{\mathcal{B}} = \{\tilde{\mathbf{x}}_1, \tilde{\mathbf{x}}_2, \cdots, \tilde{\mathbf{x}}_{2N-1}, \tilde{\mathbf{x}}_{2N}\}$ where $\tilde{\mathbf{x}}_{2k-1}$ and $\tilde{\mathbf{x}}_{2k}$ are generated by applying different transformations to each image $\mathbf{x}_k$ in the batch. The loss function for a pair of two differently-transformed images $(\tilde{\mathbf{x}}_{2k-1}, \tilde{\mathbf{x}}_{2k})$ from the input image $\mathbf{x}_k$ is defined as follows.

$$l(\tilde{\mathbf{x}}_{2k-1}, \tilde{\mathbf{x}}_{2k}) = -\log \frac{\exp\left(sim\left(f\left(\tilde{\mathbf{x}}_{2k-1};\boldsymbol{\theta}\right), f\left(\tilde{\mathbf{x}}_{2k};\boldsymbol{\theta}\right)\right)/\tau\right)}{\sum_{i=1}^{2N} \mathbb{I}[i \neq (2k-1)]\exp\left(sim\left(f\left(\tilde{\mathbf{x}}_{2k-1};\boldsymbol{\theta}\right), f\left(\tilde{\mathbf{x}}_i;\boldsymbol{\theta}\right)\right)/\tau\right)}, \quad (8)$$

where $N$ is the number of images in a batch, $f(\cdot)$ is an encoder network including a projection layer, and $sim(\mathbf{u}, \mathbf{v}) = \mathbf{u}^\top \mathbf{v}/\|\mathbf{u}\|_2\|\mathbf{v}\|_2$ is the cosine similarity. It is worth noting that we include the projection layer in the encoder network in order to obtain the transformation-invariant representations, unlike the original version of SimCLR that discards the projection layer for their downstream tasks.

In the end, the objective function of SimCLR for a batch $\mathcal{B}$ is defined as

$$\mathcal{L}_{SimCLR} = \frac{1}{2N} \sum_{k=1}^{N} \left[l\left(\tilde{\mathbf{x}}_{2k-1}, \tilde{\mathbf{x}}_{2k}\right) + l\left(\tilde{\mathbf{x}}_{2k}, \tilde{\mathbf{x}}_{2k-1}\right)\right]. \quad (9)$$

## B  IMPLEMENTATION DETAILS

We choose a 16-4 WideResnet (Zagoruyko & Komodakis, 2016) as the backbone architecture. We adopt the training strategy for multi-label classification, proposed in (Hendrycks et al., 2019). During the training, we use the cosine annealing for scheduled learning (Loshchilov & Hutter, 2016) with initial learning rate 0.1 and Nesterov momentum. The dropout rate (Srivastava et al., 2014) is set to 0.3. All the self-supervised methods based on transformation classification also use the same backbone architecture and training hyperparameters with ours. For DSVDD, we use LeNet (LeCun et al., 1998) style network as described in the paper and implementation.[2] For SimCLR, we employ the ResNet18 (He et al., 2016) with the fixed $\tau$ value of 0.5. For GROC-CL, the regularization coefficient $\nu$ for optimizing the conditional likelihoods is set to 0.0001.

## C  EXPERIMENTAL RESULTS

In Table 2, we report the full comparison results with the non-self-supervised methods for one-class classification (summarized in Figure 3).

## D  THEORETICAL BACKGROUNDS FOR GROC-CL

GROC-CL basically utilizes the encoder network $f$ that induces the latent space, where the similarity between the representation of an input image and that of each anchor transformation is modeled by an isotropic Gaussian distribution (conditioned on the transformation). Formally, the similarity between $\mathbf{x}$ and $T_j$ can be described as $sim(\mathbf{x}, T_j) = \log p(\mathbf{x}|T_j) = \log \mathcal{N}\left(f(\mathbf{x})|\boldsymbol{\mu}_j, \sigma_j^2 I\right)$, where $p(\mathbf{x}|T_j)$ is the class-conditional (or transformation-conditional) probability.

As discussed in Section 3.3, the representation of each anchor transformation should be distinguishable from the others', which is the foremost condition for GROC, in order to effectively calculate the conformity score and identify in-class/out-of-class images from the score. In this sense, GROC-CL can be understood from the perspective of Gaussian Discriminant Analysis (GDA). To this end,

---

[2]https://github.com/lukasruff/Deep-SVDD-PyTorch

we optimize the encoder network by maximizing the posterior probability of a transformed image $T_j(\mathbf{x})$ having the maximum similarity with the transformation $j$, which is denoted by $p(T_j|T_j(\mathbf{x}))$. For simplicity, we assume that the prior probability for each class (or transformation) follows the Bernoulli distribution, i.e., $p(T_j) = \beta_j / \sum_k \beta_k$.

$$
\begin{aligned}
p(T_j|T_j(\mathbf{x})) &= \frac{p(T_j)\, p(T_j(\mathbf{x})|T_j)}{\sum_{k=0}^{K-1} p(T_k)\, p(T_j(\mathbf{x})|T_k)} \\
&= \frac{\exp\left(-\left(2\sigma_j^2\right)^{-1}\|f(T_j(\mathbf{x})) - \boldsymbol{\mu}_j\|^2 - \log \sigma_j^d + \log \beta_j\right)}{\sum_{k=0}^{K-1} \exp\left(-\left(2\sigma_k^2\right)^{-1}\|f(T_j(\mathbf{x})) - \boldsymbol{\mu}_k\|^2 - \log \sigma_k^d + \log \beta_k\right)} \\
&= \frac{\exp\left(sim\left(T_j(\mathbf{x}), T_j\right) + b_j\right)}{\sum_{k=0}^{K-1} \exp\left(sim\left(T_j(\mathbf{x}), T_k\right) + b_k\right)}.
\end{aligned}
\tag{10}
$$

Note that taking the log of this equation becomes equivalent to the second term in Equation (6).

In addition, we need to force the empirical class-conditional distribution to follow the isotropic Gaussian distribution and also approximate the empirical class mean to the obtained class mean $\boldsymbol{\mu}_j$. Thus, we minimize the Kullback-Leibler (KL) divergence between the $j$-th empirical class-conditional distribution $\mathcal{P}_j$ and the corresponding Gaussian distribution $\mathcal{N}\left(\boldsymbol{\mu}_j, \sigma_j^2 I\right)$. The empirical class-conditional distribution for transformation $j$ is defined as follows.

$$
\mathcal{P}_j = \frac{1}{|\mathcal{X}_{in}^{tr}|} \sum_{\mathbf{x} \in \mathcal{X}_{in}^{tr}} \delta\left(\mathbf{z} - f\left(T_j(\mathbf{x})\right)\right),
\tag{11}
$$

where $\delta(\cdot)$ is the Dirac measure. Finally, the KL divergence is obtained by

$$
\begin{aligned}
KL\left(\mathcal{P}_j \| \mathcal{N}\left(\boldsymbol{\mu}_j, \sigma_j^2 I\right)\right) &= -\int \frac{1}{|\mathcal{X}_{in}^{tr}|} \sum_{\mathbf{x} \in \mathcal{X}_{in}^{tr}} \delta\left(\mathbf{z} - f\left(T_j(\mathbf{x})\right)\right) \log\left[\frac{1}{\left(2\pi\sigma_j^2\right)^{d/2}} \exp\left(-\frac{\|\mathbf{z} - \boldsymbol{\mu}_j\|^2}{2\sigma_j^2}\right)\right] d\mathbf{z} \\
&\quad + \int \frac{1}{|\mathcal{X}_{in}^{tr}|} \sum_{\mathbf{x} \in \mathcal{X}_{in}^{tr}} \delta\left(\mathbf{z} - f\left(T_j(\mathbf{x})\right)\right) \log\left[\frac{1}{|\mathcal{X}_{in}^{tr}|} \sum_{\mathbf{x} \in \mathcal{X}_{in}^{tr}} \delta\left(\mathbf{z} - f\left(T_j(\mathbf{x})\right)\right)\right] d\mathbf{z} \\
&= -\frac{1}{|\mathcal{X}_{in}^{tr}|} \sum_{\mathbf{x} \in \mathcal{X}_{in}^{tr}} \log\left[\frac{1}{\left(2\pi\sigma_j^2\right)^{d/2}} \exp\left(-\frac{\|f(T_j(\mathbf{x})) - \boldsymbol{\mu}_j\|^2}{2\sigma_j^2}\right)\right] + \log\frac{1}{|\mathcal{X}_{in}^{tr}|} \\
&= \frac{1}{|\mathcal{X}_{in}^{tr}|} \sum_{\mathbf{x} \in \mathcal{X}_{in}^{tr}} \left(\frac{\|f(T_j(\mathbf{x})) - \boldsymbol{\mu}_j\|^2}{2\sigma_j^2} + \log \sigma_j^d\right) + \text{constant} \\
&= -\frac{1}{|\mathcal{X}_{in}^{tr}|} \sum_{\mathbf{x} \in \mathcal{X}_{in}^{tr}} sim\left(T_j(\mathbf{x}), T_j\right) + \text{constant}.
\end{aligned}
\tag{12}
$$

The final form of the KL divergence is derived by using the definition of the Dirac measure. After the constant term is excluded, it becomes the same with the first term in Equation (6). Minimizing the KL term for all the classes (or transformations) matches the empirical class-conditional distributions with the isotropic Gaussian distributions.

Table 2: The AUROC of non-self-supervised methods for one-class classification on the three evaluation setups.

| Dataset | in-class | Fixed Viewpoint | Anchor Viewpoint | Random Viewpoint |
|---|---|---|---|---|
| | | OCSVM / DSVDD / DSVDD+ / GROC-IP/ GROC-CL | | |
| CIFAR-10 | 0 | 0.65 / 0.65 / 0.80 / 0.76 / 0.75 | 0.63 / 0.64 / 0.65 / 0.70 / 0.68 | 0.60 / 0.63 / 0.69 / 0.63 / 0.64 |
| | 1 | 0.41 / 0.54 / 0.45 / 0.95 / 0.97 | 0.38 / 0.48 / 0.45 / 0.91 / 0.94 | 0.39 / 0.46 / 0.49 / 0.84 / 0.88 |
| | 2 | 0.65 / 0.67 / 0.62 / 0.85 / 0.84 | 0.65 / 0.62 / 0.63 / 0.76 / 0.73 | 0.64 / 0.63 / 0.64 / 0.64 / 0.65 |
| | 3 | 0.50 / 0.52 / 0.54 / 0.77 / 0.77 | 0.49 / 0.52 / 0.54 / 0.73 / 0.69 | 0.49 / 0.45 / 0.53 / 0.62 / 0.61 |
| | 4 | 0.75 / 0.76 / 0.76 / 0.90 / 0.90 | 0.74 / 0.72 / 0.75 / 0.85 / 0.83 | 0.72 / 0.73 / 0.74 / 0.72 / 0.71 |
| | 5 | 0.51 / 0.51 / 0.54 / 0.88 / 0.88 | 0.50 / 0.48 / 0.49 / 0.83 / 0.83 | 0.48 / 0.45 / 0.45 / 0.72 / 0.73 |
| | 6 | 0.72 / 0.75 / 0.72 / 0.90 / 0.93 | 0.71 / 0.72 / 0.73 / 0.81 / 0.83 | 0.70 / 0.68 / 0.73 / 0.72 / 0.76 |
| | 7 | 0.51 / 0.52 / 0.58 / 0.95 / 0.96 | 0.50 / 0.51 / 0.53 / 0.91 / 0.89 | 0.50 / 0.43 / 0.54 / 0.77 / 0.77 |
| | 8 | 0.68 / 0.68 / 0.41 / 0.92 / 0.94 | 0.54 / 0.53 / 0.50 / 0.88 / 0.87 | 0.54 / 0.36 / 0.61 / 0.80 / 0.80 |
| | 9 | 0.49 / 0.52 / 0.37 / 0.90 / 0.93 | 0.38 / 0.37 / 0.39 / 0.86 / 0.88 | 0.38 / 0.40 / 0.40 / 0.79 / 0.83 |
| | avg | 0.59 / 0.61 / 0.58 / 0.88 / 0.89 | 0.55 / 0.56 / 0.57 / 0.82 / 0.82 | 0.54 / 0.52 / 0.58 / 0.73 / 0.74 |
| CIFAR-100 | 0 | 0.66 / 0.64 / 0.58 / 0.77 / 0.78 | 0.65 / 0.48 / 0.58 / 0.71 / 0.72 | 0.64 / 0.49 / 0.58 / 0.62 / 0.64 |
| | 1 | 0.52 / 0.56 / 0.47 / 0.74 / 0.72 | 0.51 / 0.57 / 0.57 / 0.66 / 0.62 | 0.49 / 0.45 / 0.54 / 0.60 / 0.62 |
| | 2 | 0.52 / 0.56 / 0.39 / 0.75 / 0.82 | 0.52 / 0.52 / 0.54 / 0.70 / 0.75 | 0.48 / 0.55 / 0.61 / 0.65 / 0.70 |
| | 3 | 0.51 / 0.58 / 0.47 / 0.80 / 0.80 | 0.50 / 0.53 / 0.57 / 0.72 / 0.75 | 0.48 / 0.53 / 0.44 / 0.52 / 0.65 |
| | 4 | 0.52 / 0.58 / 0.46 / 0.81 / 0.81 | 0.51 / 0.53 / 0.56 / 0.76 / 0.76 | 0.46 / 0.31 / 0.56 / 0.70 / 0.72 |
| | 5 | 0.44 / 0.50 / 0.43 / 0.67 / 0.66 | 0.44 / 0.50 / 0.47 / 0.60 / 0.59 | 0.43 / 0.55 / 0.51 / 0.53 / 0.55 |
| | 6 | 0.52 / 0.51 / 0.50 / 0.84 / 0.85 | 0.51 / 0.49 / 0.47 / 0.77 / 0.82 | 0.51 / 0.42 / 0.40 / 0.59 / 0.58 |
| | 7 | 0.59 / 0.63 / 0.62 / 0.67 / 0.69 | 0.59 / 0.58 / 0.55 / 0.60 / 0.62 | 0.54 / 0.56 / 0.58 / 0.59 / 0.56 |
| | 8 | 0.67 / 0.67 / 0.63 / 0.85 / 0.87 | 0.66 / 0.63 / 0.69 / 0.78 / 0.80 | 0.67 / 0.66 / 0.65 / 0.69 / 0.75 |
| | 9 | 0.69 / 0.69 / 0.51 / 0.91 / 0.91 | 0.50 / 0.52 / 0.48 / 0.89 / 0.87 | 0.55 / 0.42 / 0.47 / 0.76 / 0.77 |
| | 10 | 0.75 / 0.82 / 0.57 / 0.85 / 0.85 | 0.62 / 0.71 / 0.67 / 0.82 / 0.81 | 0.61 / 0.58 / 0.67 / 0.73 / 0.74 |
| | 11 | 0.61 / 0.61 / 0.53 / 0.86 / 0.87 | 0.61 / 0.60 / 0.58 / 0.80 / 0.81 | 0.60 / 0.54 / 0.52 / 0.70 / 0.70 |
| | 12 | 0.68 / 0.63 / 0.65 / 0.83 / 0.84 | 0.67 / 0.65 / 0.64 / 0.77 / 0.80 | 0.66 / 0.62 / 0.66 / 0.68 / 0.72 |
| | 13 | 0.59 / 0.65 / 0.66 / 0.60 / 0.59 | 0.59 / 0.60 / 0.62 / 0.56 / 0.55 | 0.58 / 0.66 / 0.65 / 0.51 / 0.52 |
| | 14 | 0.44 / 0.45 / 0.46 / 0.90 / 0.90 | 0.43 / 0.43 / 0.46 / 0.86 / 0.84 | 0.44 / 0.45 / 0.46 / 0.73 / 0.70 |
| | 15 | 0.60 / 0.65 / 0.59 / 0.71 / 0.72 | 0.60 / 0.57 / 0.59 / 0.65 / 0.65 | 0.55 / 0.58 / 0.58 / 0.60 / 0.61 |
| | 16 | 0.66 / 0.65 / 0.60 / 0.80 / 0.76 | 0.65 / 0.58 / 0.60 / 0.73 / 0.68 | 0.63 / 0.58 / 0.59 / 0.66 / 0.66 |
| | 17 | 0.71 / 0.76 / 0.70 / 0.92 / 0.92 | 0.61 / 0.48 / 0.53 / 0.88 / 0.84 | 0.57 / 0.58 / 0.61 / 0.79 / 0.76 |
| | 18 | 0.52 / 0.50 / 0.44 / 0.90 / 0.92 | 0.45 / 0.49 / 0.43 / 0.87 / 0.86 | 0.48 / 0.48 / 0.47 / 0.75 / 0.76 |
| | 19 | 0.55 / 0.56 / 0.45 / 0.87 / 0.85 | 0.49 / 0.43 / 0.44 / 0.81 / 0.79 | 0.49 / 0.47 / 0.45 / 0.73 / 0.71 |
| | avg | 0.59 / 0.61 / 0.53 / 0.80 / 0.81 | 0.56 / 0.54 / 0.55 / 0.75 / 0.75 | 0.54 / 0.53 / 0.55 / 0.66 / 0.67 |
| SVHN | 0 | 0.54 / 0.62 / 0.55 / 0.92 / 0.92 | 0.53 / 0.51 / 0.52 / 0.92 / 0.92 | 0.52 / 0.53 / 0.54 / 0.78 / 0.80 |
| | 1 | 0.51 / 0.55 / 0.50 / 0.89 / 0.90 | 0.50 / 0.51 / 0.52 / 0.89 / 0.89 | 0.50 / 0.53 / 0.52 / 0.75 / 0.77 |
| | 2 | 0.52 / 0.55 / 0.50 / 0.94 / 0.94 | 0.51 / 0.50 / 0.50 / 0.94 / 0.94 | 0.50 / 0.50 / 0.51 / 0.75 / 0.76 |
| | 3 | 0.51 / 0.53 / 0.50 / 0.93 / 0.92 | 0.50 / 0.49 / 0.50 / 0.93 / 0.91 | 0.50 / 0.50 / 0.51 / 0.75 / 0.73 |
| | 4 | 0.50 / 0.52 / 0.50 / 0.95 / 0.96 | 0.49 / 0.49 / 0.50 / 0.95 / 0.96 | 0.48 / 0.51 / 0.51 / 0.71 / 0.71 |
| | 5 | 0.52 / 0.54 / 0.52 / 0.95 / 0.94 | 0.51 / 0.49 / 0.51 / 0.95 / 0.94 | 0.51 / 0.51 / 0.51 / 0.83 / 0.78 |
| | 6 | 0.51 / 0.56 / 0.50 / 0.91 / 0.91 | 0.50 / 0.49 / 0.50 / 0.91 / 0.91 | 0.51 / 0.49 / 0.51 / 0.80 / 0.78 |
| | 7 | 0.51 / 0.56 / 0.52 / 0.96 / 0.96 | 0.50 / 0.51 / 0.52 / 0.96 / 0.96 | 0.49 / 0.51 / 0.51 / 0.70 / 0.74 |
| | 8 | 0.50 / 0.54 / 0.49 / 0.92 / 0.91 | 0.49 / 0.48 / 0.49 / 0.92 / 0.91 | 0.49 / 0.50 / 0.51 / 0.71 / 0.67 |
| | 9 | 0.52 / 0.52 / 0.51 / 0.90 / 0.89 | 0.51 / 0.49 / 0.51 / 0.90 / 0.89 | 0.51 / 0.49 / 0.52 / 0.79 / 0.78 |
| | avg | 0.51 / 0.55 / 0.51 / 0.93 / 0.92 | 0.50 / 0.50 / 0.51 / 0.93 / 0.92 | 0.50 / 0.51 / 0.52 / 0.76 / 0.75 |

