# OpenReview forum: "One-class Classification Robust to Geometric Transformation"
_ICLR.cc/2021/Conference — Reject_

### Official Review · AnonReviewer1 · 2020-10-28
**Niche paper**

**Rating:** 4
**Confidence:** 4

**Review:**

Summary: This paper considers the deep one-class classification problem. Some recent state of the art in this area is built upon self-supervised learning methods that are trained to predict the rotation applied to a training image, and then use the success of rotation prediction on test images as an outlier score. The paper observes that, while successful on standard benchmarks, this strategy is not robust to unexpected image rotations at test-time. Since, humans are (presumably) able to exhibit rotation invariance during test-time in 1-class classification, this is considered a flaw in existing methods. To rectify this flaw, the paper proposes to use an anomaly score which is the maximum over all possible rotation predictions. The results show that the proposed method outperforms prior approaches when exposed to novel rotations at test time.

Strength:
+ The paper correctly identifies a flaw in existing rotation self-supervision-based approaches to one-class deep learning. It can be seen as identifying a family of pathological examples where these methods fail.
+ The proposed modification is a reasonable, intuitive, and effective approach to correcting this flaw, and obtaining robustness to these difficult rotated examples.

Weaknesses:
1. Niche issue: This paper is ultimately focusing on a niche issue: One-class classification => Specific rotation-prediction family of one-class methods => Robustness of this specific family of methods to novel test-time rotations. It’s not clear that this issue is of sufficiently general interest to be worth publication in ICLR.
2. Practical relevance: For real applications, we would likely either work with data where photos do indeed come in a typical orientations, and thus existing methods would work fine. Or if not we would use a one-class method that aims for rotation-invariance, rather than rotation equivariance. The “bug” that this paper identifies is not surprising, and would not be likely to bite anyone in practice.
3. Technical advance/novelty: The fix proposed by this model (take the max over rotation outputs, rather than look at a single output) is rather obvious and not a significant technical advance. This modification relies on the same “low-confidence in presence of novel data” assumption that is already explored in Hendrycks ICLR’17 for class-wise probabilities.
4. Significance: The whole study is about fixing a glitch in some current high-performing self-supervision-for-1class methods. But while these methods may currently top the leaderboards for the 1class benchmarks, there are a plethora of self-supervised methods out there (jigsaw-prediction, colorisation, etc) that are yet to be adapted to 1-class learning. These may outperform the rotation-prediction methods for 1-class learning, while not being susceptible to the glitch identified here. Unless it’s clear that rotation-prediction is the final word in 1-class learning, then it may not be a significant result.

---

> ### Author Response · Authors · 2020-11-19
> **Response to Reviewer1**
>
> Thanks for your review. This is our response.
>
> First of all, we argue that data does not always come in a typical orientation. For this reason, the robustness to the viewpoint has been extensively researched and addressed in conventional image classification tasks [1, 2, 3].
>
> The main challenge that this paper tries to address (i.e., sensitivity to geometrically transformed inputs) is not limited to the existing self-supervised SOTA methods. Many other one-class classification methods also raise the same challenge or they do not make comparable results to the self-supervised methods based on the transformation prediction. For example, in our experiment, DSVDD+ and SimCLR, which are not trained for discriminating the geometric transformation, showed much worse performance and were not very effective to detect out-of-class images for both fixed/random viewpoint setups.
>
> According to some recent papers [4, 5, 6], training the one-class classifier to predict (or differentiate) geometric transformations including rotations turns out to be *the most important principle* so far, for obtaining a good performance in terms of one-class classification. For example, they showed that the self-supervised learning approach based on non-geometric transformation (e.g., sharpening, blurring, and colorization) is not effective for one-class classification. Since the self-supervised methods based on transformation-prediction consistently show SOTA performance, presenting a simple and effective way of relieving the limitation of this mainstream approach is worth it for the community. In addition, we want to emphasize that the self-supervised approach we focus on in this work is not limited to “rotation” but handles “geometric transformation” including vertical/horizontal translations as well.
>
> **References**
>
> [1] Ryu, Jongbin, Ming-Hsuan Yang, and Jongwoo Lim. "Dft-based transformation invariant pooling layer for visual classification." Proceedings of the European Conference on Computer Vision (ECCV). 2018.
>
> [2] Laptev, Dmitry, et al. "TI-POOLING: transformation-invariant pooling for feature learning in convolutional neural networks." Proceedings of the IEEE conference on computer vision and pattern recognition. 2016.
>
> [3] Shen, Xu, et al. "Transform-invariant convolutional neural networks for image classification and search." Proceedings of the 24th ACM international conference on Multimedia. 2016.
>
> [4] I. Golan and R. El-Yaniv. Deep anomaly detection using geometric transformations. In NeurIPS, pages 9758–9769, 2018.
>
> [5] L. Bergman and Y. Hoshen. Classification-based anomaly detection for general data. In ICLR, 2020.
>
> [6] D. Hendrycks, M. Mazeika, S. Kadavath, and D. Song. Using self-supervised learning can improve model robustness and uncertainty. In NeurIPS, pages 15637–15648, 2019.

---

### Official Review · AnonReviewer3 · 2020-10-29
**Good but can be improved**

**Rating:** 6
**Confidence:** 4

**Review:**

This paper presents a one-class classifier robust to geometrically-transformed inputs (GROC). A conformity score is proposed that measures how strongly an input image agrees with one of the predefined in-class transformations. Experiments show that the proposed method works well on 3 datasets for out-of-class detection and produces similar scores for in-class images under different transformations.

Overall this is a well written paper. The proposed method is well motivated and results are comprehensive. Results are quite promoting compared to existing works.

Technical:
- How the proposed method is different from SimCLR? Both try to maximize the similarity between an image under different transformations. This should be better clarified.
- A nice thing in Bergman and Hoshen's paper is that geometric transformation is generalized to the affine class, which enable it to be applied to non-visual data, such as tabular data. In this paper, the application domain seems to be confined within images only. I wonder how the proposed method can be extended to other modalities?
- While the paper mentions a challenge "to optimize the encoder network so that its outputs follows" a Gaussian distribution, it is not clear how this challenges is resolved effectively.

Experiments:
Results and analysis look good, but I do have a few concerns as follows:
- Why the performance of SimCLR is so low compared to other methods? it is somehow unexpected given that SimCLR has shown outstanding performance on unsupervised image classification as a representation learning method. Any insights?  Also, which data augmentations are you using? From SimCLR paper, the choice of data augmentations is critical, so I wonder whether this is related to the low performance here.
- Why the work: Bergman and Hoshenet al., Classification-Based Anomaly Detection for General Data is not included in experiments? It is closely related to this topic as it also applies geometric transformation for anomaly detection and compares with GT. Without such comparisons, the results are less convincing. Please provide more explanations.

---

> ### Author Response · Authors · 2020-11-19
> **Response to Reviewer3**
>
> Thanks for the kind and positive review. This is our response.
>
> **Technical**
>
> GROC aims to predict the type of geometric transformations that are applied to input images (i.e., learning transformation-discriminative representations), whereas SimCLR encourages the representations of input images geometrically-transformed from the same image to be similar to each other (i.e., learning transformation-invariant representations).
>
> In this paper, we only focus on image data. Our work could be extended to other types of data with random affine transformations as done in GOAD.
>
> We added the theoretical backgrounds for GROC-CL in the appendix, which is about how the outputs of the optimized encoder can follow a Gaussian distribution.
>
> **Experiments**
>
> At first, the set of data augmentation used for SimCLR is described in Section 4.1.
>
> One possible explanation for the low performance of SimCLR is that the property of “transformation-invariant” is overgeneralized for the out-of-class images, which are not used for training the encoder. The in-class score in Equation (7), which is designed based on the assumption that only in-class images (used for training) would make similar representations for geometric transformations, also produces the high value for out-of-class images as well; this makes it difficult to distinguish out-of-class images from in-class images, and eventually degrades the performance.
>
> According to the literature of the self-supervised one-class classification [1, 2], MGT [1] turns out to outperform GOAD [2]. We also checked that MGT beats GOAD in our pre-experiments. Furthermore, since GT, MGT, and GOAD [1, 2, 3] share the same underlying idea for their objective function and in-class scores, we do not include GOAD in our baseline methods.
>
> **References**
>
> [1] D. Hendrycks, M. Mazeika, S. Kadavath, and D. Song. Using self-supervised learning can improve model robustness and uncertainty. In NeurIPS, pages 15637–15648, 2019.
>
> [2] L. Bergman and Y. Hoshen. Classification-based anomaly detection for general data. In ICLR, 2020.
>
> [3] I. Golan and R. El-Yaniv. Deep anomaly detection using geometric transformations. In NeurIPS, pages 9758–9769, 2018.

---

### Official Review · AnonReviewer2 · 2020-10-29
**Review AnonReviewer2**

**Rating:** 5
**Confidence:** 4

**Review:**


**UPDATE**

I acknowledge that I have read the author responses as well as the other reviews. I appreciate the improvements made and clarifications given by the authors.

I would keep my score recommendation at 5 (marginally below acceptance threshold) at this point, however, mainly due to (i) a limited novelty (as partly acknowledged by the authors in the response) and (ii) a limited experimental evaluation (missing high-resolution datasets such as ImageNet or real-world AD datasets such as MVTec-AD) of the current manuscript.

I encourage the authors to build on their current results and extend their work with additional datasets and an analysis of geometric shifts also at training time.

#####


**Summary**

This work considers the unsupervised anomaly detection task on image data. The paper critically remarks that existing state-of-the-art self-supervised methods [5, 6, 2], which learn one-class models through auxiliary classification based on applying a set of geometric transformations, are sensitive to changes in viewpoint due to a one-to-one identification of auxiliary labels with certain transformations, thus assuming a fixed viewpoint for inlier data. A Geometrically Robust One-class Classifier (GROC) is proposed which is a variant of previous self-supervised methods [5, 6] where at testing time, instead of the one-to-one relation between transformations and labels, an inlier conformity score is derived from the most confident in-class transformation agreement for a given input. This enables, even when there is a geometric change of viewpoint for some images at test time, to find the best matching in-class transformation, thus implicitly arranging an image with the most characteristic in-class viewpoint learned from the training data. The sum of such conformity scores obtained from applying the set of transformations to an input finally serves as an overall inlier score. Two ways to measure agreement in the output layer are suggested, dot product similarity and conditional likelihood, which show similar detection and robustness performance. Experiments using the one-vs-rest evaluation protocol on CIFAR-10, CIFAR-100, and SVHN demonstrate that GROC indeed is more robust to geometric changes, which is simulated by three experimental scenarios that cover various degrees of changes in viewpoint at testing time.


**Pros**

+ The proposed method, GROC, presents a simple and elegant tweak to make self-supervised anomaly detection methods more robust to geometric changes and shifts at testing time.
+ The paper is technically correct and the presented experimental evaluation scientifically rigorous.
+ The paper has a clear and easy to follow structure.
+ The work is placed well into the existing literature.


**Cons**

- The novelty of the presented work is rather low as the non-robustness of self-supervised methods to geometric changes is not surprising, given that these models have been trained on the premise that the training data has similar viewpoints and geometry. The methodological novelty is also rather low, as GROC presents a test time score adaptation, but the training of self-supervised models is the same as in [5, 6, 2].
- The experimental evaluation is limited to low-resolution synthetic anomaly detection settings and somewhat tailored to fit the proposed model as geometric shifts and changes are only applied to the testing data in the proposed evaluation scenarios, but GROC still relies on the criticized similar viewpoint assumption for model training.


**Recommendation**

I tend towards rejecting this paper (score: 5) as I find the novelty of the presented work to be rather low and the current experiments insufficient for evaluating model robustness to geometric transformations.

The presented method, GROC, still relies on the assumption that the viewpoint in the training data is mostly the same, given that training a model follows [5, 6, 2] and the training data remains untransformed. In my opinion, the experimental evaluation should also consider a geometric augmentation of the training data to evaluate the robustness. In fact, such training data augmentation is used with the non-self-supervised methods (DSVDD+ and SimCLR) to make them more robust to geometric variation, but this is also the harder setting to learn from. GROC should be evaluated on such a setting as well to properly assess its robustness. Variation in viewpoints in both, the training and test data, is also the more relevant and natural setting in practice. Such an analysis would be interesting and could be insightful for the community to assess the usefulness of self-supervised methods based on geometric transformation in general.


**Additional feedback and ideas for improvement**

- Include geometric variation in the training data into the experimental evaluation.
- Include higher resolution datasets with more geometric variation (e.g., ImageNet [6, 8]) and real-world anomaly detection datasets (e.g., MVTec [3]) into the analysis.
- Is GROC beneficial on non-image data as well, as considered by GOAD [2]?
- Other previous work has also shown that self-supervised methods are vulnerable to non-geometric anomalies [4].
- The one-vs-all evaluation scheme has seen some critique recently [1, 7]. It would be good to consider leave-one-out and real anomaly detection datasets (e.g., MVTec [3]) as well.


**Minor Comments**

1. Section 1 : ‘Nevertheless, their supervision is not useful enough to capture the semantic of high-dimensional data for a target class, which eventually leads to *a* limited performance.’ Citation? Also ‘supervision’ might be misleading here, as the referred to methods are unsupervised. Maybe ‘learning’?
2. Section 1, 2nd paragraph: GOAD [2] also can be applied to general data, e.g. tabular data. Not only images.
3. Section 2.2: ‘*For self-supervised* learning, [...]’
4. Section 2.2: ‘Using the self-labeled dataset, *these methods* train a softmax classifier [...]’
5. Insufficient space below Figure 1 caption.
6. Figure 1: Should be $S_{in}(x')$ and $S_{in}(x'')$, correct?
7. Section 3.3: ‘[...], can be defined in various *ways*.’
8. Section 3.3.1: ‘After we build a softmax classifier by adding the linear classification layer of weights $W$ *on top* of the encoder network [...]’ There is no change in architecture to previous methods, right? [5, 6] employ linear layers before the cross-entropy as well, correct? The difference lies in taking the maximum over dot products/logit values at testing time?
9. Section 4.1: ‘OCSVM is a classical kernel-based method for one-class classification, which finds a *maximum-margin hyperplane the separates* most of the training in-class examples.’
10. Section 4.4.1: ‘[...], the classification-based methods cannot beat *random guessing*, [...]’
11. Section 4.4.1: ‘In conclusion, both *of* our methods [...]’
12. Section 5: ‘[...], whereas the state-of-the-art methods perform even worse than *random guessing*.’


#####

**References**

[1] F. Ahmed and A. Courville. Detecting semantic anomalies. In AAAI, pages 3154–3162, 2020.

[2] L. Bergman and Y. Hoshen. Classification-based anomaly detection for general data. In ICLR, 2020.

[3] P. Bergmann, M. Fauser, D. Sattlegger, and C. Steger. Mvtec ad–a comprehensive real-world dataset for unsupervised anomaly detection. In CVPR, pages 9592–9600, 2019.

[4] P. Chong, L. Ruff, M. Kloft, and A. Binder. Simple and effective prevention of mode collapse in deep one-class classification. In IJCNN, pages 1–9. IEEE, 2020.

[5] I. Golan and R. El-Yaniv. Deep anomaly detection using geometric transformations. In NeurIPS, pages 9758–9769, 2018.

[6] D. Hendrycks, M. Mazeika, S. Kadavath, and D. Song. Using self-supervised learning can improve model robustness and uncertainty. In NeurIPS, pages 15637–15648, 2019.

[7] L. Ruff, J. R. Kauffmann, R. A. Vandermeulen, G. Montavon, W. Samek, M. Kloft, T. G. Dietterich, and K.-R. Müller. A unifying review of deep and shallow anomaly detection. arXiv preprint arXiv:2009.11732, 2020.

[8] L. Ruff, R. A. Vandermeulen, B. J. Franks, K.-R. Müller, and M. Kloft. Rethinking assumptions in deep anomaly detection. arXiv preprint arXiv:2006.00339, 2020.

---

> ### Author Response · Authors · 2020-11-19
> **Response to Reviewer2**
>
> Thank you for your sincere review and insightful comments. This is our response.
>
> **Ideas for Improvements**
>
> As you pointed out, it seems obvious that the existing self-supervised methods are not robust to geometric changes at all, as their models have been trained on the assumption that training/test images have similar viewpoints and geometry. However, there have not been any attempts to address this challenge before in terms of one-class classification, whereas the robustness to the viewpoint has been extensively researched and addressed in conventional image classification tasks [1, 2, 3]. In this sense, our work has a novelty to some extent.
>
> In this paper, we present a novel test time score adaptation that can effectively relieve the sensitivity to geometric transformations, while keeping the self-supervised learning strategy during the training. Despite the simpleness of the method (though someone might criticize it as not-novel), GROC successfully achieves the best performance for more advanced (random-viewpoint) setups, without compromising the performance for the original (fixed-viewpoint) setup.
>
> In addition, the main challenge of one-class classification tasks (including anomaly detection, outlier detection, and novelty detection) is that no one can know the test distribution during the training phase (i.e., only in-class images are given for training the classifier). The goal of these tasks is to detect the test samples that differ from the training samples in some regards. From this perspective, the consideration of the case that geometric shifts and changes are only applied to the testing data does make sense. Note that the training data can be refined to have homogeneous properties (e.g., the fixed viewpoint) in many applications, unlike the testing data that we are not able to control at all. We argue that training a robust and powerful one-class classifier using the images having diverse viewpoints is a totally different problem, and at the same time, we agree that this problem also should be addressed. For this reason, we leave this for our future work.
>
> Furthermore, the most recent work on anomaly detection [4] defined several types of test samples that are likely to be considered as out-of-class in real-world applications, in order to cover more practical scenarios: 1) “non-semantic distributional shifts” refers to the test samples that keep semantic of training samples but have slightly different visual factors (e.g., viewpoint or geometry), and 2) “semantic distributional shifts” means the test samples that belong to novel object classes. Similar to our work, [4] also argued that only “semantic distributional shifts” should be flagged down as out-of-class while “non-semantic distributional shift” should be considered as in-class.
>
> **Additional Responses**
>
> We will include higher resolution datasets in the future.
>
> The claim about non-geometric anomalies presented in [5] does not conflict with ours. They pointed out that the existing self-supervised one-class classifiers are likely to identify “non-geometrically transformed in-class images” as in-class, under the assumption that such images should be detected as out-of-class. On the contrary, our paper assumes that both geometrically/non-geometrically transformed in-class images are still in-class. Thus, according to [5], it is obvious that the methods are not vulnerable to non-geometrically transformed inputs. For this reason, we focus on geometric transformation more than non-geometric transformation in our work.
>
> We edited our paper considering your comments in the minor comments section.
>
> **References**
>
> [1] Ryu, Jongbin, Ming-Hsuan Yang, and Jongwoo Lim. "Dft-based transformation invariant pooling layer for visual classification." Proceedings of the European Conference on Computer Vision (ECCV). 2018.
>
> [2] Laptev, Dmitry, et al. "TI-POOLING: transformation-invariant pooling for feature learning in convolutional neural networks." Proceedings of the IEEE conference on computer vision and pattern recognition. 2016.
>
> [3] Shen, Xu, et al. "Transform-invariant convolutional neural networks for image classification and search." Proceedings of the 24th ACM international conference on Multimedia. 2016.
>
> [4] F. Ahmed and A. Courville. Detecting semantic anomalies. In AAAI, pages 3154–3162, 2020.
>
> [5] P. Chong, L. Ruff, M. Kloft, and A. Binder. Simple and effective prevention of mode collapse in deep one-class classification. In IJCNN, pages 1–9. IEEE, 2020.

---

### Official Review · AnonReviewer4 · 2020-11-01

**Rating:** 4
**Confidence:** 3

**Review:**

In this paper, the authors aim to solve the problem of one class classification using self-supervision. While, this method has been adopted previously, in this paper, the authors aim to make the one-class classification robust to rotations.

The main idea in the method is based on the idea of using anchor transformations instead of augmenting the dataset using transformed examples. The method is compared against other self-supervised one class classification methods on CIFAR-10/CIFAR-100 and SVHN datasets

The novelty in the proposed work is limited as it specifically addresses the issue of geometric transformations for one-class classification and it is a very specific case that is addressed. The techniques adopted are also fairly straightforward. In the first technique, K anchor transformations are used to obtain the prediction with the learned self-supervised method. The second technique obtains the conditional likelihood conditioned on each transformation. These techniques are not particularly novel.

The evaluation is limited as the paper argues that data-augmentation would not be applicable as it would result in inconsistent supervision. However, no actual evaluation is provided for the same. Methods like SIM-CLR advocate the use of contrastive learning with many classes. It is not evident that using the same approach for one-class classification can be done using just a single class. It would be interesting to consider additional evaluation where SIM-CLR is evaluated only with the additional one-class that is of concern.

---
I have considered the rebuttal provided. Particularly the aspect that data-augmentation would result in inconsistent supervision is an interesting point and experimental analysis of the same would be useful. However, I am not convinced that the paper provides a broad enough solution. In view of this I raise my score from 3 to 4, but maintain my view that the paper is presently not good enough for acceptance.

---

> ### Author Response · Authors · 2020-11-19
> **Response to Reviewer4**
>
> Thanks for your review. This is our response.
>
> **Concerns about Novelty**
>
> There have not been any attempts to address our problem before (i.e., the robustness to the viewpoint) in terms of one-class classification, whereas it has been extensively researched and addressed in conventional image classification tasks [1, 2, 3]. In this sense, our work has a novelty to some extent.
> In this paper, we present a test time score adaptation that can effectively relieve the sensitivity to geometric transformations, while keeping the self-supervised learning strategy during the training. Despite the simpleness of the method (though someone might criticize it as not novel), GROC successfully achieves the best performance for more advanced (random-viewpoint) setups, without compromising the performance for the original (fixed-viewpoint) setup.
>
> **Concerns about Evaluation**
>
> In order to show that data-augmentation cannot be applicable to the self-supervised one-class classifiers (i.e., GT and MGT), we tried to train them in our pre-experiment. However, we finally failed to train the classifier by using the augmented dataset thus did not include the result in the paper. Intuitively, it is obvious that the dataset augmented by geometric transformation results in inconsistent supervision for the self-supervised one-class classifier, which should be able to discriminate the geometric transformations (please refer to Figure 1).
>
> In addition, as you pointed out, SimCLR was not proposed for one-class classification but for contrastive learning with many classes. However, regardless of one-class/multi-classes, the goal of SimCLR is to make the representations of geometrically-transformed images similar to each other. In this sense, it can be inherently robust to various viewpoints (or geometric transformation), thus we considered it as an additional baseline method. In the end, as shown in our result, SimCLR cannot achieve the performance as high as transformation-discriminative methods for both the fixed/random-viewpoint setups.
>
> **References**
>
> [1] Ryu, Jongbin, Ming-Hsuan Yang, and Jongwoo Lim. "Dft-based transformation invariant pooling layer for visual classification." Proceedings of the European Conference on Computer Vision (ECCV). 2018.
>
> [2] Laptev, Dmitry, et al. "TI-POOLING: transformation-invariant pooling for feature learning in convolutional neural networks." Proceedings of the IEEE conference on computer vision and pattern recognition. 2016.
>
> [3] Shen, Xu, et al. "Transform-invariant convolutional neural networks for image classification and search." Proceedings of the 24th ACM international conference on Multimedia. 2016.

---

### Decision · Program_Chairs · 2021-01-07
**Final Decision**

**Decision:**

Reject

**Comment:**

This paper explores the robustness of one-class classifiers to geometric transformations at test time. The authors observe that some existing methods fail to detect novel images from the same class when they have undergone specific transformations at test time i.e. in-plane rotations. In contrast, it is suggested that humans have no difficulty in ignoring the impact of these types of transformations. To address this issue, the authors propose to take the maximum prediction over the set of rotated versions of a given test image.

The current consensus from reviewers, and this meta-reviewer agrees with this view, is that the paper, while not without some merit, is too narrow in focus to be of general interest in its current form. The main contribution is limited to one family of transformations, and it is not immediately clear how to generalize this to others when the entire transformation space is not easily enumerated. There are also legitimate concerns regarding if the specific issue outlined is likely to be a problem in practice (see R1's comments). The authors allude to some interesting negative results related to data augmentation in their response to R4 (R2 also had questions about this). The authors should consider adding these results to a future revision of the paper as it will strengthen the central message.

In conclusion given the limited support, this AC also agrees that the paper is not yet ready for publication at ICLR.